# The Rare Condition of a Double Cervix: Results from the High-Risk Human Papillomavirus-Based Cervical Cancer Screening Program in the Lazio Region

**DOI:** 10.3390/v16071149

**Published:** 2024-07-17

**Authors:** Tiziana Pisani, Ettore Domenico Capoluongo, Maria Cenci

**Affiliations:** 1Unità Operativa Complessa di Patologia Clinica, Azienda Ospedaliera, San Giovanni-Addolorata, 00184 Rome, Italy; tpisani@hsangiovanni.roma.it (T.P.); mcenci@hsangiovanni.roma.it (M.C.); 2Dipartimento di Medicina Molecolare e Biotecnologie Mediche, Università Federico II, 80138 Napoli, Italy

**Keywords:** HPV, double cervix, endometrial cancer screening

## Abstract

Precancerous and cancerous lesions of the uterine cervix are known to be associated with Human Papillomavirus (HPV) infection. The screening of high-risk (HR)-HPV infection in the female population has led to the discovery of several cases of a double cervix, a congenital malformation that is very rare. The purpose of this study was to evaluate HR-HPV infections in women with a double cervix within the National Cervical Cancer Screening program of the Lazio region (Italy). From June 2021 to March 2024, a total of 142,437 samples were analyzed by Seegene’s Anyplex TM II HR-HPV method, which identifies 14 HR-HPV genotypes. For each woman identified with a double cervix, two separate samples were taken from both cervices and analyzed separately. Twenty-seven women with a double cervix were identified (0.019%): 23 women were tested as negative for both cervices, while the remaining four (namely A, B, C, and D) resulted positive. By genotyping, the following results were obtained: (A) Both samples showed genotype 31; (B) one cervix was negative while the other showed genotype 58; (C) one cervix was positive for HPV 18 and 31 while for 18, 31, and 33 in the other; and (D) one cervix showed genotype 66 while the other carried the 66 and 68 genotypes. Double cervix is a very rare condition where the presence of HR-HPV genotypes is not homogeneous. As already described, our study confirms that different genotypes can be detected in double cervix malformation, suggesting the need to perform HPV screening on brushing samples from both cervices.

## 1. Introduction

HPV genital infections are quite frequent in a woman’s life; although they are mostly transient, these infections can cause cervical dysplasia or neoplasia, particularly when associated with high-risk (HR) HPV [1].

Cervical carcinoma represents the most frequent neoplasm of the female genital tract, and after breast, colon, and lung cancer, it is the most frequent cause of illness and death from cancer [2,3]. To prevent cervical cancer, the Lazio region (Italy), in agreement with the National Cancer Screening program, offers HR-HPV tests to all women aged 30–64 years old. During the sampling phase, rare cases of genital malformations, such as double cervices, have been detected. This uterine malformation is due to non-fusion of the Müllerian duct system during embryogenesis and can be associated with various genitourinary system malformations [4].

The aim of this work is to evaluate HR-HPV infections in women with double cervices discovered within the HR-HPV-based Cervical Cancer Screening program of the Lazio region (Italy).

## 2. Methods

In our observational descriptive study, we report data collected from June 2021 to March 2024: a total of 142,437 samples collected from women aged 30–64 years were analyzed. Samples were collected using the cervix brush and diluted in ThinPrepR PreservCyt Solution (Hologic, Marlborough, MA, USA). 

For each woman identified with a double cervix, two separate samples were taken from both cervices, and these samples were analyzed separately. 

The DNA extraction, PCR amplification, and extended genotyping were performed using an automated DNA extraction and PCR setup platform STARlet IVD (Seegene, Seoul, Republic of Korea). The Anyplex™ II HR-HPV Detection kit (Seegene, Seoul, Republic of Korea) was applied for HPV DNA detection and genotyping using the CFX96™ Real-time PCR System (Bio-Rad, Berkeley, CA, USA), as already reported [5]. Our molecular platform (the Seegene’s proprietary DPO™ and TOCE^TM^ technologies, Seoul, Republic of Korea) allows the detection and genotyping of the following 14 HR-HPV subtypes 16, 18, 31, 33, 35, 39, 45, 51, 52, 56, 58, 59, 66, and 68 [5]. All personnel (consisting of three technicians and two medical doctors) working in the specific HPV-LAB was fully dedicated to this screening, ensuring the complete standardization of the entire pipeline. 

In the Lazio region, the computerized transmission of results was carried out using SIPSO 2.0 software: our laboratory is the first to apply this informatic tool to manage the overall procedures surrounding HPV screening. The flexibility of the SIPSO software allowed us to accept separately the two samples collected from the double cervix (the same individual with two cervix brushes) on the SIPSO 2.0 software so that we could report the HR-HPV results separately, sometimes different. In fact, this software tool does not consider the rare events as double cervices. 

The HPV-positive cases were sent to a cytologic laboratory for the pap test, according to the Lazio region algorithms.

## 3. Results and Discussion

### 3.1. Results

In total, 27 women (32–64 years old; median age of 45.6 years) with a double cervix were identified (0.019%): 23 women (85.2%) tested negative for both cervices, while the remaining four (namely A, B, C, and D) (14.8%) resulted as positive (Table 1).

In one woman (A), both cervix samples showed genotype 31 (Figure 1); in the second woman (B), the left cervix sample was negative, while the right sample showed genotype 58; in the third woman (C), the right cervix sample tested positive for HPV 18 and 31, while the left sample for 18, 31, and 33 (Figure 1). In the last woman (D), one cervix showed genotype 66, while the other tested positive for HPV 66 and 68 (Figure 1).

In the three cases (women B, C, and D) showing different results on the two brush cervix samples, we repeated the HR-HPV extended genotyping. A repeated HPV test confirmed all the previous results. Regarding the differences in the eight peaks depicted in Figure 1C, we cannot exclude that the anatomy of each cervix, the cellularity recovered, and the level of infection by each genotype can be related to the number of cells guesting the individual genotypes.

The cytologic diagnoses of the positive HR-HPV cases are reported in Table 1.

### 3.2. Discussion

It is known that the main risk factor for development of cervical cancer is HR-HPV infection; therefore, it is preventable and can be easily treated if detected at early stages [6]. To prevent cervical neoplastic pathologies, the Lazio region, according to the national indications, structured a screening program involving a large number of women [5]. The gynecological examination followed by cervix brush performed on a large number of women allowed to identify women with genital malformations such as the double cervix.

This congenital malformation is estimated to have an incidence close to 0.2–0.4% in the general population and a prevalence of around 4–7% [7,8,9]. Likewise, dysplastic and neoplastic pathologies of the cervix in patients with this malformation are extremely rare, while only a few similar findings are described in the literature [4,8] in relation to HPV screening.

It is interesting to note that, although several authors described cases of carcinomas involving both cervices [8,9,10], where the detected neoplasia was present only in one of the cervices [11], other authors reported different degrees of dysplastic/malignant lesions in the two cervices [12,13]. Most of these cases were symptomatic and diagnosed only when the women complained of genital bleeding or abnormal pain [4].

Fox et al. [14], Loo et al. [15], and Sparic et al. [16] found that the pap test of these patients showed cytological atypia, such dyskaryosis and koilocytosis, suggestive of HPV infection.

Only a few authors published data related to HR-HPV genotypes: Zong et al. [12] reported two HPV-16-positive cervical carcinoma cases; Wang et al. [4] described the case of a woman with squamous cell carcinoma affecting both cervices who was HPV-16/18 positive; and Pinto et al. [17] reported a case of a double cervix with bilateral high squamous intraepithelial lesion (HSIL) associated in the right cervix with HR-HPV-33 genotype and in the left cervix with HR-HPV-35 genotype. Possible explanations for the different findings could be the different viral loads in the two cervices and the different development of epithelia and squamocolumnar junctions.

To our knowledge, there are no works in the literature regarding cervical carcinoma screening program of women with congenital malformations such as a double cervix.

The aim of screening campaigns is to monitor the persistence of HR-HPV in order to early diagnose and treat dysplastic lesions before they evolve into an infiltrating neoplasm [5].

Our data show that 14.8% of the women with double cervices resulted as HR-HPV positives; this percentage is slightly higher than the overall population examined (13.9%) [5], in our laboratory. Unfortunately, data on sexual behaviors, immune status, and social conditions of these women are not available.

Two of four HPV-positive cases showed discordant genotypes among double cervices; this difference could be due to either a low viral load or the contamination occurring during brush sampling with material from the other cervix (C and D females, Figure 1). In female B, only one of the two cervices was positive for HPV (Figure 1). It is interesting to note that the two cervices can be infected by different viral genotypes. Our findings confirm data already reported in a single case [17], also considering that the neoplasm can affect both cervices with lesions of the same or different grade [10,17,18] or only one of the two cervices [11]. We can speculate that, in women with this malformation, the two cervices may be differently sensitive to the HR-HPV infection with different risks of developing neoplasia. Our data show that cytological findings were the same on both cervices; nevertheless, the only one low-grade squamous intraepithelial lesion (LSIL) case was associated with multiple HR-HPV infections [19]. Finally, a possible limitation of our work is the lack of a detailed description of the abnormalities found beyond the cervix, as we are only able to know that there are two cervices without any further information regarding the level of anomalies. 

## 4. Conclusions

Our data are in keeping with the literature and confirm that uterine malformations are rare events and can remain unrecognized for several years. The national cervical cancer screening programs in these particular cases are important not only for the early diagnosis and treatment of neoplastic lesions but also to identify these rare abnormalities. Since the two cervices may be differently susceptible to the viral infection, it is necessary to evaluate and analyze them separately to be conclusive in reporting potential risks of infection and cancer. Obviously, these twenty-seven double cervix women were identified thanks to the non-self HPV sampling procedure; therefore, we can only underline that the cervix brush under standardized conditions can not only improve the quality of analytical HPV results but also better characterize patients status. In this regard, a recent publication has suggested that the vaginal self-collection conditions need to be modified to optimize sample recovery and performance in cervical cancer screening [20].

## Figures and Tables

**Figure 1 viruses-16-01149-f001:**
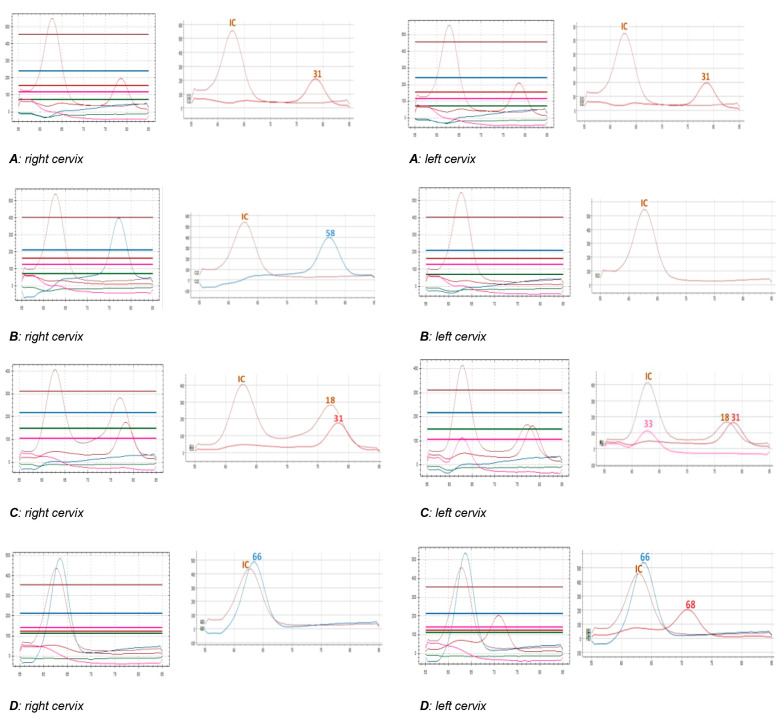
Raw data and test results on Seegene Viewer for the right and left cervices of (**A**–**D**) females. IC: internal control.

**Table 1 viruses-16-01149-t001:** Genotyping obtained on the four double cervix females positive for HR-HPV.

Female	Age (years)	HR-HPV Right Cervix	HR-HPV Left Cervix	Pap Test Right Cervix	Pap Test Left Cervix
A	32	31	31	Negative	Negative
B	53	58	/	Negative	Negative
C	54	18,31	18,31,33	LSIL	LSIL
D	52	66	66,68	Negative	Negative

LSIL: low squamous intraepithelial lesion.

## Data Availability

Data are contained within the article.

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
