# Peer review of "The Rare Condition of a Double Cervix: Results from the High-Risk Human Papillomavirus-Based Cervical Cancer Screening Program in the Lazio Region"

_viruses, 2024, doi:10.3390/v16071149_

Round 1
Reviewer 1 Report
Comments and Suggestions for Authors
General comment:
This is an interesting and important brief report, but if considered to be accepted, some need to be revised and extended as following suggestions. Besides, this article also needs to correct English wording and grammar.
Major suggestions:
1. If there's suspicion of a double cervix of the uterus, ultrasound or MRI are usually the preferred imaging methods for confirming the diagnosis. Ultrasound is commonly chosen initially because it's non-invasive and offers real-time imaging. However, MRI might provide more detailed images, especially if complexities exist or further characterization of the anomaly is necessary. CT scans aren't typically the primary choice for assessing uterine anomalies due to radiation exposure and their limitations in imaging soft tissue structures like the cervix and uterus compared to ultrasound or MRI.
How do you confirm the “Double cervix” from your study?
2. How do you avoid the intra-observer errors and inter-observer errors? Is the same doctor or technician doing the HPV DNA test in a total of 142,437 samples from 14 June 2021 to March 2024?
3. In Abstract:
A. “Twenty-seven women with double cervix were identified (0.019%): 20
women were tested as negative for both cervices, while the remaining four
(namely A, B, C, and D) resulted positive.” The total case numbers are not
correct. How about the other 3 women?
B, The (HR) HPV should appear after the first high-risk human
papillomavirus in your paper.
4. Methods:
Is this article a retrospective study? Single center or multiple centers? Where is the IRB number or ethics committee? What is the inclusion or exclusion criteria?
5. In Discussion:
A. On page 4, Line 110, write cervix should be changed to “right cervix”.
B. What is the possible cause that your study has a higher incidence of HPV positive rate in the double cervix than the overall population examined?
C. In Line 127, tow cervices should change to “two cervices”.
D. In Figure 1 (C), Why is the peak of the HPV 18 curve higher than the HPV 31 curve in the right cervix, but almost the same height of the peak between the HPV 18 and HPV 31 in the left cervix? What is the possible mechanism that induced this result? Please add to the discussion.
E. Can the Doctor or technician who collects the different samples use the same force and the same scope of the sample? Because if not the same force or scope, the amount of specimen taken will vary.

This article needs to correct English wording and grammar.
Author Response
Suggestions:
- If there's suspicion of a double cervix of the uterus, ultrasound or MRI are usually the preferred imaging methods for confirming the diagnosis. Ultrasound is commonly chosen initially because it's non-invasive and offers real-time imaging. However, MRI might provide more detailed images, especially if complexities exist or further characterization of the anomaly is necessary. CT scans aren't typically the primary choice for assessing uterine anomalies due to radiation exposure and their limitations in imaging soft tissue structures like the cervix and uterus compared to ultrasound or MRI.
How do you confirm the “Double cervix” from your study?
Response: The regional protocol provides women with medical examination by an expert gynecologist who performed cervix brush under standard conditions. Since this is a regional screening offered free of charge to all women at risk, it is unlike that the physical examination and the cervix brush sampling were also complemented with a MRI/CT or a deeper radiological investigation, because the evidence of double cervix war obtained during the physical examination. Due to the privacy rules protecting these women, we cannot have any access to other possible laboratory or radiological tests. We can only ensure that the gynecologist sent to our LAB two different samples labelled as left or right cervix. Nevertheless, we assume that the gynecologist have suggested to these women to refer to a specialized center to better characterize the anomalies found. However, we have added a sentence within the discussion where we state that: “a possible limitations of our work is the lack of a detailed description of the abnormalities found beyond the cervix as we are only able to know that there are two cervix, without any further information regarding the level of anomalies”
- How do you avoid the intra-observer errors and inter-observer errors? Is the same doctor or technician doing the HPV DNA test in a total of 142,437 samples from 14 June 2021 to March 2024?
Response: We have a dedicated staff covering the section of HPV screening. All professionals (three technicians and 2 MD) were certified for their skills and are under continuous upgrading of their professional skills on the specific setting of HPV-screening. Moreover, our platform is completely automated and therefore the errors due to personnel are absent: the two MD supervise the technicians and are able to monitor realtime (thanks to our HPV infrastructure) the status of any run. We have realized as within the period indicated there were no differences in the results obtained in relationship to the personnel working in the HPV-lab. Our system is also able to check the level of preservative liquid present in any container: this allow us to immediately identify mixed samples or containers that have not been carefully closed with subsequent loss of the ThinPrepR PreservCyt Solution (happened only for 1 sample). We also have in each run one NEG SAMPLE + 3 POS control samples, along with third party controls (1 pos and 1 neg, as already reported in the reference 5). To better describe this, we have added the following sentence within M&M section: “All personnel (consisting of three technicians and two medical doctors) working in the specific HPV-LAB is fully dedicated to this type of test, ensuring the complete standardization of the entire pipeline”
In Abstract:
- “Twenty-seven women with double cervix were identified (0.019%): 20
women were tested as negative for both cervices, while the remaining four
(namely A, B, C, and D) resulted positive.” The total case numbers are not
correct. How about the other 3 women?
B, The (HR) HPV should appear after the first high-risk human
papillomavirus in your paper
Response.
Thanks a lot for these suggestions. We have amended the text as requested.
Methods:
Is this article a retrospective study? Single center or multiple centers? Where is the IRB number or ethics committee? What is the inclusion or exclusion criteria?
Response: This is a single center an observational descriptive study where we only report aggregated data coming from our routine HPV assays in the context of regional screening. The health regional system of Lazio region allowed this screening and each gynecologist submit the informed consent to patients where they accept that their data can be used for epidemiological purposes. Therefore, every patient attending the ambulatory of each local hospital, accepted to participate to this free of charge HPV screening, where they also accepted to be contacted, in presence of a positive result, for follow-up, further investigations and treatments. Therefore, this is an epidemiological overview of the HPV positivity in the contest of a very rare condition as the double cervix.
In Discussion:
- On page 4, Line 110, write cervix should be changed to “right cervix”.
- done
- What is the possible cause that your study has a higher incidence of HPV positive rate in the double cervix than the overall population examined?
- we cannot explain this because, due to the privacy, we cannot have more information regarding sexual behaviors, immune status and social conditions of these women. Nevertheless, our group already published (ref 19) data on multiple infections where “In total, 4,244 (13.94%) were positive: 3,290 (77.52%) showed a single genotype infection and 954 (22.48%) an infection with 2 to 5 different genotypes. In 721 (75.60%) cases, two different genotypes were detected, in 191 (20.00%) there were three genotypes, in 41 (4.30%) cases there were four genotypes and in only one case (0.10%) five different genotypes were detected. HPV 16 (262 cases of co-infections) was associated in 27 cases with HPV 31 genotype, in 25 cases with HPV 68 and in 18 cases with HPV 58 “. Therefore the data on double cervix are completely in agreement with those found on the population screened.
- In Line 127, tow cervices should change to “two cervices”.
- done
- In Figure 1 (C), Why is the peak of the HPV 18 curve higher than the HPV 31 curve in the right cervix, but almost the same height of the peak between the HPV 18 and HPV 31 in the left cervix? What is the possible mechanism that induced this result? Please add to the discussion.
- we can only speculate that the genotype load within the two cervices depends on the cellularity recovered and on the microenvironment. Moreover, we cannot exclude that the single anatomy of the cervix and the level of infection by each genotype can be related to the number of cells guesting the individual genotype. We can only assure that these genotypes were present. We have added a specific sentence in the result section.
- Can the Doctor or technician who collects the different samples use the same force and the same scope of the sample? Because if not the same force or scope, the amount of specimen taken will vary.
Response: we cannot exclude any possible variability in the sampling, although we have reported as all gynecologists are fully dedicated to this regional screening and we have to assume that they are really skilled on the topic. All samples arriving to are lab are qualified for the analysis since all internal control passed the cut-off related to the cellularity.
English has been reviewed as requested.

Reviewer 2 Report
Comments and Suggestions for Authors
The description of cases with double uterine cervix has been little explored in the literature. The findings described here are of great interest as they demonstrate the difference between distinct HPV genotypes in the cervix. The complexity of the event challenges our knowledge regarding how we should proceed in these cases, which, although rare, represent an additional challenge in the observation of these women.
Author Response
WE thank the reviewer for the positive comments
Reviewer 3 Report
Comments and Suggestions for Authors
This interesting short communication describes the findings of HPV molecular cervical screening and subsequent cytology findings in individuals with double cervices variants in an Italian Provence screening scheme. Τhe relevant literature is scant, perhaps because of the rarity of this situation.
Discussion, Line 93:
Despite the absence of a wide consensus in categorization, the authors might wish to give an insight on the breakdown of their cases in the main different Mullerian categories.
Discussion, Line 100 & Lines 127-129:
Most HPV genotypes show varying tropism for different lower genital tract localizations (cervix, vagina, etc). Possible explanations for the different findings could be the different viral loads in the two cervices, the different development of epithelia and squamocolumnar junctions.
Discussion, Lines 114-115:
Self sampling strategies have globally become increasingly popular following the COVID pandemic. A brief comment would be meaningful on the feasibility of performing vaginal self sampling in women with mullerian variants. Could vaginal self sampling be recommended or considered altogether in women with double cervices, or will these situations continue to remain unrecognized with self sampling without the aid of the vigilant clinician’s eye??
Overall, an interesting manuscript worth considering for publication.
Comments on the Quality of English Language
There is room for improvement regarding Language polishing.
Author Response
Discussion, Line 93:
Despite the absence of a wide consensus in categorization, the authors might wish to give an insight on the breakdown of their cases in the main different Mullerian categories.
Response: Unfortunately, these data are not available and are out of the scope of this study.
Discussion, Line 100 & Lines 127-129:
Most HPV genotypes show varying tropism for different lower genital tract localizations (cervix, vagina, etc). Possible explanations for the different findings could be the different viral loads in the two cervices, the different development of epithelia and squamocolumnar junctions.
Response: Thanks a lot for this suggestion, that we have now included in the discussion.
Discussion, Lines 114-115:
Self sampling strategies have globally become increasingly popular following the COVID pandemic. A brief comment would be meaningful on the feasibility of performing vaginal self sampling in women with mullerian variants. Could vaginal self sampling be recommended or considered altogether in women with double cervices, or will these situations continue to remain unrecognized with self sampling without the aid of the vigilant clinician’s eye??
Response: Thanks a lot for this suggestion, that we have now included in the conclusion section
Overall, an interesting manuscript worth considering for publication
Thanks for this positive comment.
